# Community Dynamics and Engagement Strategies in Establishing Demographic Development and Environmental Surveillance Systems: A Multi-Site Report from India

**DOI:** 10.3390/healthcare11030411

**Published:** 2023-01-31

**Authors:** Nandini Sharma, Subrata Kumar Palo, Devi Madhavi Bhimarasetty, Kesava Lakshmi Prasad Kandipudi, Anil J. Purty, Tivendra Kumar, Saurav Basu, Alice Alice, A. Velavan, Sathish Madhavan, Temsunaro Rongsen-Chandola, Narendra Kumar Arora, Shikha Dixit, Sanghamitra Pati, Shikha Taneja Malik

**Affiliations:** 1Department of Community Medicine, Maulana Azad Medical College, New Delhi 110002, India; 2Regional Medical Research Centre, Bhubaneshwar 751023, India; 3Department of Community Medicine, Rangaraya Medical College Kakinada, Kakinada 533001, India; 4Department of Community Medicine, Guntur Medical College, Guntur 522004, India; 5Pondicherry Institute of Medical Sciences, Puducherry 605014, India; 6Centre for Health Research and Development, Society for Applied Studies, New Delhi 110016, India; 7Public Health Foundation of India, Gurugram 122002, India; 8INCLEN-Mawphlang DDESS, East Khasi Hills District, Shillong 793108, India; 9The INCLEN Trust International, New Delhi 110020, India; 10Department of Biotechnology, National Biopharma Mission, Biotechnology Industry Research Assistance Council (BIRAC), New Delhi 110003, India

**Keywords:** global health, community health, demography, community engagement, population health

## Abstract

Background: Six diverse Demographic Development and Environmental Surveillance System (DDESS) sites were established in urban slum, urban resettlement, peri-urban, rural, and tribal areas located in Northern, North-East, Eastern, and Southern regions of India from June 2020 to March 2022. Understanding the community dynamics and engaging people in the community is critically important in the process of establishing DDESS. We ascertained the barriers, challenges, and facilitators during the establishment of multiple DDESS sites across India. Methods: This was a cross-sectional descriptive mixed-methods study. Results: Multiple barriers and challenges encountered were reported in the process of community engagement (CE), such as geographical inaccessibility, language barriers, adverse weather, non-responsiveness due to perceived lack of individual benefit or financial gain, fear of contracting COVID-19, COVID-19 vaccine hesitancy, etc. Facilitators in the CE process were pre-existing links with the community, constitution of community advisory boards, community need assessment, concomitant delivery of outreach health services, and skill-building facilities. Conclusion: Most community barriers in the development of DDESS sites in resource-limited settings can be overcome through a multipronged approach, including effective community engagement by focusing on demonstrating trust at the local level, enlisting community mobilization and support, utilizing pre-existing community linkages, initiating community diagnosis, and meeting perceived community health needs.

## 1. Introduction

Demographic Development and Environmental Surveillance Systems (DDESS) enable the development of novel, innovative, and real-world solutions for achieving healthier, prosperous, and sustainable communities in resource-limited settings [1,2,3]. In these settings, there is a lack of information about demography- and health-related events, which limits the understanding of factors influencing health outcomes and related health inequalities required for health-system strengthening [4,5,6]. The establishment of demographic surveillance systems through longitudinal data collection is useful for identifying the existing gaps and trends in various health and development indicators in the resident communities, which could be generalized to communities with similar socio-demographic profiles [7,8]. DDESS also provide a platform to carry out site-specific research in anthropological, biomedical, and demographic parameters, as well as providing opportunities to impart training for health personnel [9]. The limited health information and vital statistics data due to the suboptimal functioning of existing civil registration and health information in most low- and middle-income countries (LMICs) is bridged by these sites [10].

Globally, there are over thirty demographic surveillance sites across Asia, Africa, and America [7]. For obtaining population-based health information, India, with a huge population size of 1.43 billion, relies primarily on “national surveys” such as Census Surveys, District-Level Household Surveys, Civil Registration Systems, the National Family Health Survey (NFHS), Sample Registration System data, and “individual studies” from established demographic surveillance sites. Nationally, there are only a few population-based surveillance sites, such as the HDSS-Ballabgarh, Vadu HDSS-Pune, and DDESS Palwal [11,12]. This low number of surveillance sites is not sufficient to provide vital information, leading to detrimental impacts such as poor estimates of burden of disease/health conditions and planning policy interventions consequently, weakening the country’s public health system [11,13,14,15]. To offset these gaps, the National Biopharma Mission—Biotechnology Industry Research Assistance Council (NBM BIRAC) supported the creation of six different demographic surveillance sites (SOMAARTH DDESS) across India (Figure 1), consisting of vulnerable populations from urban, rural, and tribal areas using a common implementation protocol. SOMAARTH is a cross-sectoral approach aiming to promote health and prevent, control, and mitigate diseases through the involvement of multiple sectors, such as Rural and Urban Development, Women and Child Development, Environmental Pollution, Agriculture, and other industries, such as Food Processing and Human Resource Development, through community engagement [1].

According to the Centre for Disease Control (CDC), community engagement (CE) is the process of working collaboratively with and through groups of people affiliated by geographic proximity, special interest, or similar situations to address issues affecting the well-being of those people [16]. Local community support and building trust through CE is essential for establishing DDESS, which facilitates the validated and prospective data collection from the community. Conceptualization and implementation of a robust protocol for improved CE enables comprehension of the community dynamics and execution of development initiatives and public health programs in these areas [13,17,18]. Such CE initiatives may include interpersonal communication, counselling sessions, and leveraging the existing cadre of front-line health workers [19]. Furthermore, CE also enables the achievement of need-based health assessment and community diagnosis, along with mainstreaming health promotion and disease prevention activities, thereby strengthening prevention components of the national health programs [20].

There is a paucity of evidence towards understanding the real-world strategies and solutions that enable effective CE in public health and developmental research [4]. Moreover, since DDESS can be utilized prospectively for conducting community-based drug and vaccine trials, understanding the means of addressing community concerns and ethical hazards by using CE processes in the resident populations becomes significant [4,13]. Consequently, this report describes the community dynamics, barriers, and community-engagement-driven enablers in the establishment of multiple geographically and demographically diverse DDESS sites across India.

## 2. Methods

Study Design: This was a cross-sectional descriptive mixed-methods study. Baseline data were collected from the ongoing multi-site DDESS project in India.

Study Setting: Six diverse DDESS sites across India established during the period from June 2020 to March 2022 were included (Figure 1).

All these sites were pre-existing field practice areas of the concerned medical colleges and research organizations. At each site, data were collected on the type of settlement (rural/urban slum/urban resettlement/etc.), demographic, and health characteristics from both DDESS surveys and non-DDESS secondary data sources, such as previous local surveys, census data, community and facility-based research studies conducted in the area, and medical and health records from the local government primary care facilities. 

Description of the study sites (DDESS): 

The six DDESS sites were slated to be developed by six respective institutions: MAMC, Gokulpuri, New Delhi; CHRD-SAS, Dakshinpuri, New Delhi; INCLEN, Mawphlang and Sohiong, Meghalaya; ICMR-RMRC, Bhubaneswar, Tigiria, Cuttack, Odisha; AMC, Vishakapatnam, Andra Pradesh; and PIMS, Marakanam-Villupuram district, Chitamur-Chengalpet district, Tamilnadu, and Muthialpet in U.T. Puducherry). The basic sociodemographic characteristics of the study population are reported in Table 1.

The Gokulpuri DDESS is an urban low-income neighbourhood in the North-East district of Delhi comprising an urban resettlement colony, urban slum, and a village. Of the total population, 91.6% are literate, 94.94% belong to the Hindu religion, and 52.28% belong to the scheduled caste. The settlement has very few (i.e., 4) health care facilities, while the nearest referral health facility is around 6–7 km away.The Dakshinpuri DDESS is an urban low-income neighbourhood located in the Southern district of Delhi. A total of 99.55% of the population is urban and 78.91% follow the Hindu religion, followed by the Muslim and Sikh religions. The settlement has 3 healthcare facilities for local residents.The Mawphlang DDESS spreads across the 364.48 Sq Km area in the Mawphlang and Sohiong administrative blocks of the East Khasi Hills district of Meghalaya state. The site is predominantly occupied by the Khasi tribal community (99.5%). In total, 80.7% of the population are Christians and the site has very high literacy rates (82.8%). There are 2 community health centres, 1 primary health centre and 9 sub-centres.The Tigiria DDESS is a predominantly rural area (89.2%). A total of 95.17% of people belong to the Hindu religion, followed by other minority communities. The majority belong to the upper caste (83.4%), followed by the schedule caste (13.2%) and the schedule tribe (3.4%). Cultivation is the main occupation. While the site has 2 community health centres, 4 primary health centres and 13 sub-centres, the referral health centre is around 15–20 km away.The Simhachalam DDESS is located in the western part of Visakhapatnam and is predominantly peri-urban. A total of 95.9% of the population belong to the Hindu religion. The scheduled caste (7.6%) and scheduled tribe (14.3%) constitute one fourth of the population. Occupations are mostly service or running small businesses. The site has the fair availability of 92 hospitals in the urban setting and 13 hospitals in the rural setting, respectively. The Marakanam, Chitamur, and Muthialpet DDESS is predominantly rural, with the average literacy rate being 65.2%. The major occupations are agriculture and fishing. The site has an availability of 48 hospitals in the rural setting, compared to only 2 hospitals in the urban facilities for the residents.

**Table 1 healthcare-11-00411-t001:** Sociodemographic characteristics of the study population.

	SAS	MAMC	AMC	RMRC	PIMS	INCLEN
Sociodemographic Characteristics *	N = 55,445	N = 54,614	N = 55,529	N = 76,379	N = 56,025	N = 85,123
Gender-wise distribution						
Male	28,513 (51.43%)	28,951 (53.01%)	27,482 (49.5%)	40,026 (52.40%)	27,571 (49.21%)	41,659 (48.9%)
Female	26,909 (48.53%)	25,661 (46.99%)	28,045 (50.5%)	36,349 (47.59%)	28,452 (50.78%)	43,464 (51.1%)
Others (3rd Gender)	23 (0.04%)	2 (0.01%)	2(0%)	4 (0.01%)	2 (0.00%)	0
Age-wise distribution						
Up to 5 yrs	4205 (0.69%)	4650 (8.51%)	3601 (6.4%)	5772 (7.56%)	4144 (7.40%)	13,999 (16.5%)
6–14 yrs	7937 (4.85%)	8213 (15.04%)	7026 (12.6%)	9802 (12.83%)	7361 (13.14%)	20,212 (23.7%)
15–45 yrs	32,787 (58.1%)	31,050 (56.85%)	30,198 (54.3%)	38,772 (50.76%)	29,337 (52.36%)	40,931 (48.1%)
46–59 yrs	6240 (19.2%)	6472 (11.85%)	9450 (17%)	11,891 (15.57%)	8286 (14.79%)	6041 (7.1%)
More than equal to 60 yrs	4276 (17.2%)	4229 (7.74%)	5254 (9.4%)	10,142 (13.28%)	6897 (12.31%)	3940 (4.6%)
Religion-wise distribution *						
Hindu	49,013 (88.4%)	51,857 (94.95%)	52,378 (94.3%)	72,493 (94.91%)	51,223 (91.43%)	49 (0.06%)
Muslim	4700 (8.5%)	2417 (4.43%)	977 (1.8%)	2629 (3.44%)	2962 (5.29%)	14 (0.02%)
Christian	925 (1.7%)	84 (0.15%)	1894 (3.4%)	25 (0.03%)	1657 (2.96%)	68,693 (80.7%)
Jain	42 (0.1%)	95 (0.17%)	13 (0%)	21 (0.03%)	4 (0.01%)	0
Buddhist	51 (0.1%)	19 (0.3%)	4 (0%)	1205 (1.58%)	0	0
Sikh	609 (1.1%)	119 (0.22%)	6 (0%)	6 (0.01%)	0	34 (0.04%)
Others					180 (0.32%)	Khasi 15,301 (17.9%)
Caste-wise distribution						
General	16,296 (29.4%)	19,655 (35.99%)	9267 (17%)	16,855 (22.07%)	1155 (2.06%)	36 (0.04%)
Scheduled Caste (SC)	31,140 (56.2%)	19,748 (36.16%)	7265 (13%)	8224 (10.77%)	7104 (12.68%)	129 (0.1%)
Scheduled Tribe (ST)	348 (0.6%)	1192 (2.18%)	1165 (2%)	5382 (7.05%)	2103 (3.75%)	84,922 (99.7%)
Other Backward Caste (OBC)	5321 (9.6%)	13,230 (24.22%)	37,693 (68%)	45,878 (60.07%)	42,757 (76.32%)	27 (0.03%)
Marital status						
Married	26,185 (47.2%)	26,474 (48.47%)	34,146 (61.4%)	46,610 (61.02%)	33,508 (59.81%)	26,630 (31.2%)
Unmarried	25,849 (46.6%)	25,444 (45.59%)	21,383 (38.5%)	29,787 (39%)	22,517 (40.19%)	52,869 (62.2%)
Literacy status						
Literate	45,895 (82.8%)	47,735 (87.40%)	43,361 (78.08%)	65,755 (86.09%)	51,342 (91.64%)	70,301 (82.8%)
Illiterate	4608 (8.31%)	4558 (8.35%)	12,168 (21.92%)	10,624 (13.38%)	4683 (8.36%)	6423 (7.6%)

* Not added to 100% total due to missing data.

Sample size and sampling strategy: A universal sample (census) of all the households located in the DDESS sites were included in this study through a standardized house-to-house survey process. 

Data collection (quantitative) was conducted following a systematic process followed for the establishment of comprehensive community-based surveillance sites [12]. Broadly, site establishment activities were divided into two parts, i.e., (1) understanding the local context and establishing the administrative architecture, which included seeking permission from the local stakeholders, setting up site office space, building a multidisciplinary team, and engaging local people in the data collection processes; and (2) establishing the data collection system, which included the translation and piloting of DDESS survey tools and procedures, GIS mapping and census enumerations, quality assurance, and data harmonization. Data gathered at these sites were the synthesized and disseminated information of maps and census factsheets given to the village councils and other stakeholders. The survey collected key information on built-up areas, such as family composition, physical and mental disabilities, chronic diseases, health-seeking behaviour, socio-economic status, drinking water sources, and environmental hygiene, as well as details on non-residential activities. This process has been previously validated in an existing DDESS site in India with a high degree of certitude [13]. Furthermore, experiences were documented at the site level through a series of interviews with key project staff and field investigators from each of the DDESS sites who were involved in the planning, development, and implementation of the CE strategies. 

Data entry: All 6 DDESS sites used Android-based electronic questionnaires called ‘SOMAARTH 1 & SOMAARTH 3’. The enumerator asked the respondent questions in a face-to-face interview and the responses were recorded by the enumerator directly into their data collection tablet. Data were securely saved in the Amazon Web Service (AWS©) server. 

Data analysis: Data were downloaded from the server and cleaned using STATA Version 15 (STATACorp, College Station, TX, USA) and IBM SPSS Statistics for Windows, Version 25.0. (Armonk, NY, USA: IBM Corp). Data related to key sociodemographic characteristics of the study populations were reported as frequency and percentages.

Written transcripts were prepared for the interviews as part of this study. Thematic analysis was used for identification of recurrent themes, all of which had been identified a priori from a comprehensive review of the literature. Key findings from each of the sites were centrally analysed using qualitative synthesis techniques, such as textual narrative and framework synthesis for the identification of the key barriers, facilitators, and coping strategies in community engagement during the establishment of DDESS sites. 

Analytical conceptual framework: The CE activities undertaken by the study teams were classified within a conceptual framework comprising the previously validated five stages of the CE continuum [15,18], which include: a. informing the community members; b. consulting the community representatives (administrators, frontline workers, community advisory board members); c. involving the local community residents (village volunteers, women’s self-help groups); d. collaborating with local health departments and non-governmental organizations; and e. empowering the community for health promotion and disease prevention through awareness programs. The process adopted and activities carried out are presented in Figure 2.

## 3. Results

The locations of six DDESS sites included in this study are presented in Figure 1. The observational findings are presented under four thematic sections: A. demography and community dynamics; B. barriers encountered during CE; C. enablers to achieve CE; and D. strategies adopted for better CE. Demography and community dynamics: The geographic locations of the study sites are unique in nature, showing variation in terms of urban, peri-urban, rural, and tribal-predominant areas. People were residing in plain, riverine, hilly terrain, and forest areas depending on the location of the DDESS sites. The site-specific demographic information is reported in Table 1. 

(A)Barriers encountered during community engagement (CE). The significance of CE exists both during the establishment of the DDESS and ensuing quality data collection from the sites. The research teams encountered many barriers and challenges in the process of ensuring adequate CE. While some common challenges were observed in multiple sites, some were site-specific:
Limited or lack of understanding of locally spoken languages: India is the world’s most linguistically diverse country. Absence of familiarity with the locally spoken language and dialect posed challenges for investigators and project staff, especially in remote and tribal regions with predominantly indigenous populations. For instance, Khasi, an Austroasiastic language, is predominantly spoken in the tribal areas of Mawphlang and Sohiong blocks of the North-Eastern state of Meghalaya, having just over 1 million speakers. This caused difficulty for site investigators unfamiliar with the local language to communicate with members of the local community that precluded effective CE and the data collection processes.Poor road connectivity and difficult terrain: Some DDESS sites were located in areas (rural and tribal) that were not connected with all-weather motorable roads and caused difficult access. The survey team had to travel considerable distances by foot to reach these remote villages. The interior areas located in forest and hilly terrain were difficult to reach, and more so during the rainy season.Inclement weather and extreme climatic conditions: While DDESS sites in Meghalaya were the wettest, the site of Odisha is near the riverine regions. Such sites are prone to flood and difficult to carry out field activities in during the rainy seasons. The temperature extremities (summer in southern DDESS sites and winter in the northern part) and heavy rainfall were also detrimental to conducting routine project activities.Expectation for financial benefits: Some community members had expectations of financial gain or compensation for participating in or facilitating the project and research activities in their sites. In the absence of provision for any direct financial benefit, such members were disinclined to participate or consent.Fear of wage loss: Some family members, especially the adults who were mainly involved in earning livelihoods for their family, could not afford to spend time with the team during their working hours.Migration: Some people migrate in search of jobs, even with their family. This causes a loss in follow-up of the registered population and a challenge to DDESS sustainability.Dispute over land ownership: Households with prevailing legal disputes about their land ownership were initially found to be resentful and reluctant to share any of their personal information as part of the study. In hilly areas, even the village councils are involved in disputes over the village boundaries, and some of them do not agree with the inter-village boundaries given in the government records.Pandemic and fear of COVID-19 infection: The COVID-19 pandemic created a lot of panic among the community, and the imposed restrictions created hindrance for initiating interactions with outsiders in the sites. The unavailability of hospital beds for COVID patients during the COVID peak resulted in non-cooperation of the community with the research team.Distrust for COVID-19 vaccination: Communities, particularly rural and tribal communities, were reluctant to accept COVID-19 vaccination. They were panic-stricken and agitated at the prospect of engaging with people working in the health sector or associated with hospitals. There was also resistance among communities against the implementation of non-pharmaceutical measures, such as social distancing and following COVID norms. Dissatisfaction with suboptimal functioning of primary care facilities during the pandemic, reduced staffing, and services as a result of the pandemic were a major issue faced by the surveillance team.Unpleasant past experiences: Some of the community members had unpleasant experience of supporting governmental or non-governmental activities in the past, resulting in non-cooperation. For instance, no personal benefits were received by the community members for previous participation in other surveillance activities. Some community members had unpleasant experiences of not receiving any financial or other benefits promised to them for their participation in prior community-based research activities conducted in their areas, which further compounded their unwillingness to participate in the present research project.Engagement of locally hired people in the enumeration team may bring undue demands and interference from the local community, such as mounting pressure to hire undeserving candidates, negotiations for the payment amount or rolling back the termination order, etc.(B)Enablers to achieve CE: Many enabling factors were noticed that facilitated CE, which are described below.
Pre-existing community links with the research teams by virtue of being the field practice areas of the respective institutions; b. broadcasts of the activities conducted by the SOMAARTH DDESS among the public by the village council through the public announcement system; c. involvement and engagement of community-level frontline health workers (ASHA, Anganwadi workers); and d. involvement of village volunteers were helpful in ensuring community engagement activities.(C)Strategies adopted for better community engagement: In view of the existing barriers, challenges and enablers, various site-specific strategies were undertaken in order to achieve better community engagement (Table 2).
Formation of a community advisory board (CAB) at each DDESS site: Each study site formed a CAB by identifying and recruiting significant stakeholders following a common CAB charter for better functioning of the DDESS site. Meetings among the members of the advisory committee were held to identify and address existing community-level issues to achieve trust and better community involvement and participation. The CAB helps in supporting advocacy, sensitization campaigns related to the project, and mitigation of any conflict, while ensuring that beliefs and cultural disparities among community members are respected. The CAB gives advice and recommendations to the research team but has no administrative or legal responsibilities. The research team informs the CAB before carrying out any new activities. The board provides guidance and assistance in planning and implementing the activities of the project by ensuring that the activities are conducted in an informed and transparent manner that is acceptable to the community.Identifying and involving community volunteers: Some of the DDESS sites identified and involved local community volunteers, such as Panchayati Raj Institution (PRI) members, schoolteachers, ASHA, Anganwadi workers, etc. These community volunteers were engaged during the community mobilization process. They provided enormous support during community consultation, village boundary mapping, and for health awareness initiatives towards strengthening surveillance activity. In order to avoid the interference of the community in the recruitment and research processes, a detailed SOP for the village volunteers’ recruitment was developed and followed by the sites. The formal recruitment process for the engagement of village volunteers included vacancy announcements in all the site villages/localities, organization of written examinations and tablet tests at clusters for ensuring the participation of volunteers from the villages which are far from the office location, followed by face-to-face interviews with the successful candidates. A merit list was prepared and publicly displayed for the transparency of the process and recruitment of the village volunteers was carried out on the basis of the merit list.Community consultations involving key stakeholders: Prior to data collection, the population was consulted and appraised of the project objectives. During the meetings, key stakeholders, such as local health care providers, schoolteachers, and community leaders were also involved. Many stakeholders played a role as a link between the community and researchers. This improved the trust among the people to participate and support the project’s activities.Undertaking other health-related activities: Apart from the project-related activities, the DDESS sites also organized some other health initiatives, such as health camps, COVID-19 Serosurveys, screening for non-communicable diseases (NCDs), and oral health problems. Community awareness and sensitization programs based on prevailing key health issues, such as the COVID-19 pandemic situation, prenatal/postnatal care, multidrug-resistant tuberculosis (MDR-TB), and vector-borne diseases, such as dengue, chikungunya, etc., were supported by the community stakeholders.The participatory mapping and data collection approach helped in gathering updated relevant information in the manner in which the community perceived them. In villages where boundaries were not available in the government census maps, research teams were able to gather information through the community consultation process.


**Table 2 healthcare-11-00411-t002:** DDESS site wise comparison of the barriers, enablers and strategies for achieving enhanced community engagement.

Factors	MAMC, New Delhi	SAS, New Delhi	INCLEN, Shillong	RMRC, Bhubaneswar	AMC, Andhra Pradesh	PIMS, Puducherry
Barriers encountered						
Language barrier	No	No	Yes	Yes	No	No
Poor road connectivity and difficult terrain	No	No	Yes	No	No	No
Inclement weather and extreme climate conditions	No	No	Yes	Yes	Yes	Yes
Expectation of financial benefits	No	Yes	Yes	Yes	Yes	Yes
Fear of wage loss	Yes	Yes	Yes	Yes	Yes	Yes
Migration	Yes	Yes	Yes	Yes	Yes	Yes
Dispute over land ownership	Yes	Yes	No	Yes	Yes	Yes
Pandemic and fear of COVID-19 infection	Yes	Yes	Yes	Yes	Yes	Yes
Distrust of COVID-19 vaccination	No	Yes	Yes	Yes	Yes	Yes
Unpleasant past experiences	No	No	Yes	Yes	No	Yes
Community interference	No	No	Yes	Yes	No	No
Enablers						
Pre-existing community links	Yes	Yes	Yes	Yes	Yes	Yes
Broadcast of activities	No	No	Yes	No	No	Yes
Involvement and engagement of frontline health workers	Yes	Yes	Yes	Yes	Yes	Yes
Involvement of village volunteers	Yes	Yes	Yes	Yes	Yes	Yes
Strategies adopted						
Formation of community advisory board (CAB)	Yes	Yes	Yes	No	Yes	Yes
Identifying and involving community volunteers	Yes	Yes	Yes	Yes	Yes	Yes
Community consultations involving key stakeholders	Yes	Yes	Yes	Yes	Yes	Yes
Undertaking other health-related activities (screening camps, serosurveys, etc.)	Yes	No	No	Yes	Yes	Yes
Participatory mapping	Yes	Yes	Yes	Yes	Yes	Yes
Village Volunteer recruitment SOP	Yes	Yes	Yes	Yes	Yes	Yes
Communication strategy	Yes	Yes	Yes	Yes	Yes	Yes

The barriers and enablers encountered for CE, and the strategies adopted to achieve better CE, are depicted as a conceptual framework (Figure 3).

A comparison od the barriers and enablers encountered, and strategies adopted for better community engagement, across the six DDESS sites, is presented in Table 2.

## 4. Discussion

The establishment of DDESS is important to study a population cohort through collecting data longitudinally and monitoring their health and wellbeing. This helps to understanding the community dynamics, the existing challenges, and barriers better to developing appropriate disease-control and prevention strategies. These sites enable epidemiological surveillance through prospective data collection on population dynamics and health measures in a specific geographical area, generating data on a country’s vital statistics related to disease burden and demographic information that instructs and advances public health policy formation [21].

There is growing global recognition of the role of CE in facilitating and accentuating research activities planned and conducted within communities. However, there is a lack of global consensus guidelines on assessing the quality of CE processes [22]. Furthermore, the COVID-19 pandemic has signified the role of CE strategies in supporting community interventions for risk reduction, community surveillance, and enhancing COVID-appropriate behaviours in populations, especially in the resource-constrained settings of the developing world [23]. Finally, the processes followed for achieving CE in DDESS sites can yield valuable opportunities towards demonstrating community trust, enlisting community resources, and realizing effective communication for improved community response and participation [24]. 

In this study, all the participating organizations had been functioning in their respective sites for >10 years and were engaged in various public health services, and this activity helped the research team to acquire knowledge and understand the beliefs and socio-cultural practices of the local population. Operational challenges for the sustained functioning of the DDESS sites are evident from the requirement of frequent updating of population data prospectively often resulting in research fatigue amongst respondents [25]. Inaccessibility of households due to safety or privacy concerns or lack of trust have been previously reported during the development of HDSS sites in South Africa [26,27]. Anticipating these challenges in advance, most of the sites during the investigation prioritized the recruitment of local field investigators, conducted health and screening camps, and planned for the provision of continued health interventions to the community. As an important lesson, the recruitment of local field investigators (village volunteers) should be carried out following a strict recruitment SOP with transparency. Mitigation strategies to reduce non-responses included inculcating the teachings during pre-testing, wherein an enhanced focus was placed on sensitizing field investigators to carefully approach and justify questions related to income, occupation, identity, and household possessions. Additionally, the donning of culturally appropriate attire, and substituting some working hours during evenings or on holidays to allow the interviewing of working male household members, further increased community responsiveness. Specific challenges experienced by field investigators in ensuring community participation included privacy concerns arising from the requirement for disclosure of personal and familial information, including wealth, perceived lack of individual benefit, and some dissatisfaction from suboptimal delivery of essential health services during the COVID-19 pandemic. Our findings suggest that applying CE-related strategies for addressing these concerns and visualizing new opportunities to benefit the community were effective in achieving the project’s goals. It is important for research teams to give a true and realistic picture of what can be delivered and never heighten expectations—unrealistic expectations cannot be delivered, and if commitments are not fulfilled the community will become suspicious. This delays the processes of building trust and confidence in the research team. Initially (when a new team comes to a village), community expectations are very high—the community members may keep mentioning these for the first few months, but these need to be watered down to realism in a tactful but consistent manner to avoid later challenges [28].

Some DDESS sites are being developed amongst vulnerable communities, such as tribals, rural, and urban poor, necessitating the need for community consultation to identify community concerns which may influence community participation. In this regard, Community Advisory Boards (CABs) were frequently used to support CE by helping research teams honour the community values, and frequently also guide decision-making when deliberating on the choice of community-based research and supportive activities [27,28,29]. All sites set up CABs and held frequent meetings, sensitising them before starting any activity. However, a pertinent challenge in CE is the need for authentic and inclusive community representation, enabling their participation. Consequently, in this project, the appointment of members to the Community Advisory Boards was undertaken, mindful of the context of the need for maintenance of diverse social representation and avoiding any selection bias favouring highly empowered individuals, which is a known threat to such a model of engagement [16]. The engagement of the community in the mapping and data collection processes became an intervention, as they built trust among community members, developed the capacities of the field investigators, and produced a sense of community pride [30].

Furthermore, the suboptimal functioning of existing health surveillance systems in the more geographically inaccessible remote, rural, and tribal locations renders the monitoring and detecting of the vital events and associated data management difficult [31,32]. All the study sites ensured prioritization of mobilization of the local frontline workers, particularly the ASHAs, to identify health needs and gaps in the target communities. Our findings corroborate evidence supportive of mobilizing frontline workers as a key strategy for increasing long-term capacity building and responsiveness in detecting and responding to any emergent health threats with concurrent health promotion in DDESSs functioning in resource-limited settings [33]. 

Effective community mobilization and collaboration with motivated community partners are also crucial towards identifying the prevalent social determinants of health that impact population health [1,6]. In the present project, community partners included the recruitment of locals as field investigators, incentivization models for frontline and community health workers, partnerships with local NGOs and schoolteachers who were leveraged for identifying local perspectives on community health-related problems, and likely solutions apart from promoting confidence and trust between the community and the research teams [1,4].

## 5. Conclusions

The establishment of DDESS sites intends to bridge the health- and demography-related data gaps necessary for informing and strengthening public health policies in developing countries. The specific challenges encountered in establishing DDESS sites in different areas, and the approaches and strategies implemented to overcome these, may be instructive in developing comparable platforms under similar conditions. Most community barriers exist in the development of DDESS sites in resource-limited settings, but most can be overcome through effective community engagement by focusing on enlisting local community trust, mobilization and support, utilizing pre-existing community linkages, community diagnosis, and meeting health needs. The study underscores the need for rigorous community mobilisation and regular stakeholder participation, for a better understanding of community dynamics, and aiding community engagement. 

## Figures and Tables

**Figure 1 healthcare-11-00411-f001:**
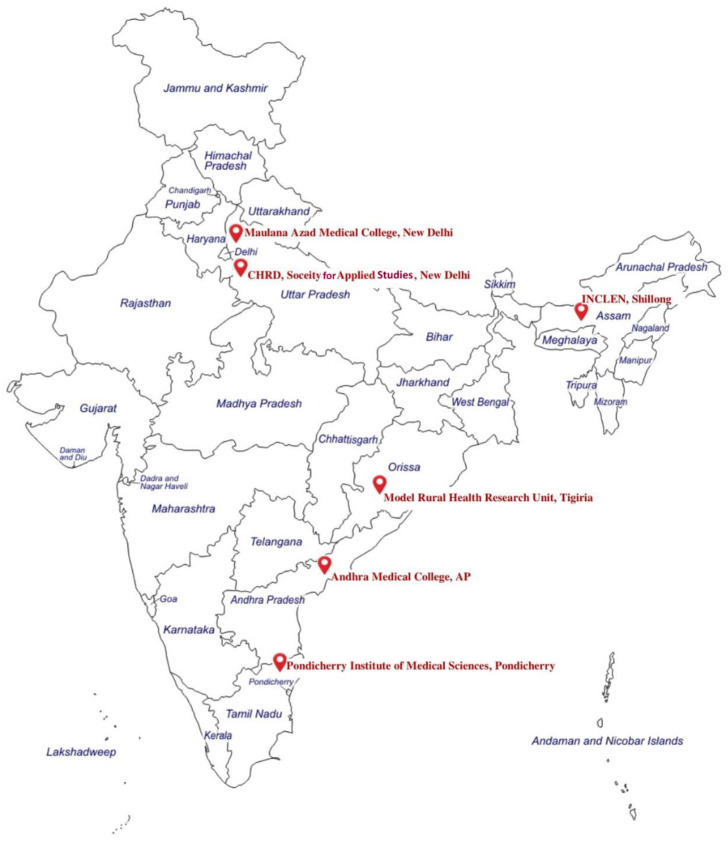
Locations of the planned Demographic Development and Environmental Surveillance System (DDESS) project sites in India.

**Figure 2 healthcare-11-00411-f002:**
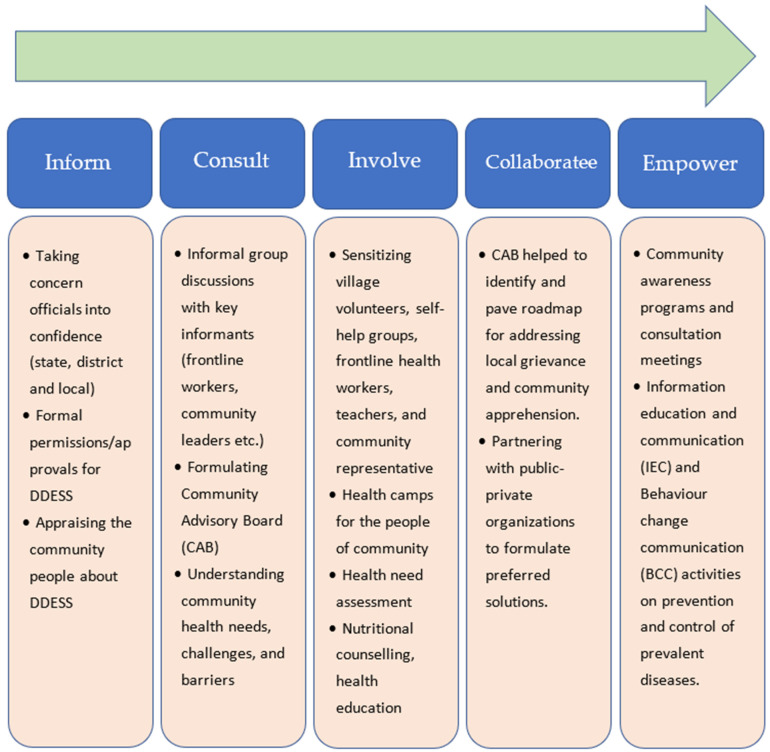
Process adopted for community engagement.

**Figure 3 healthcare-11-00411-f003:**
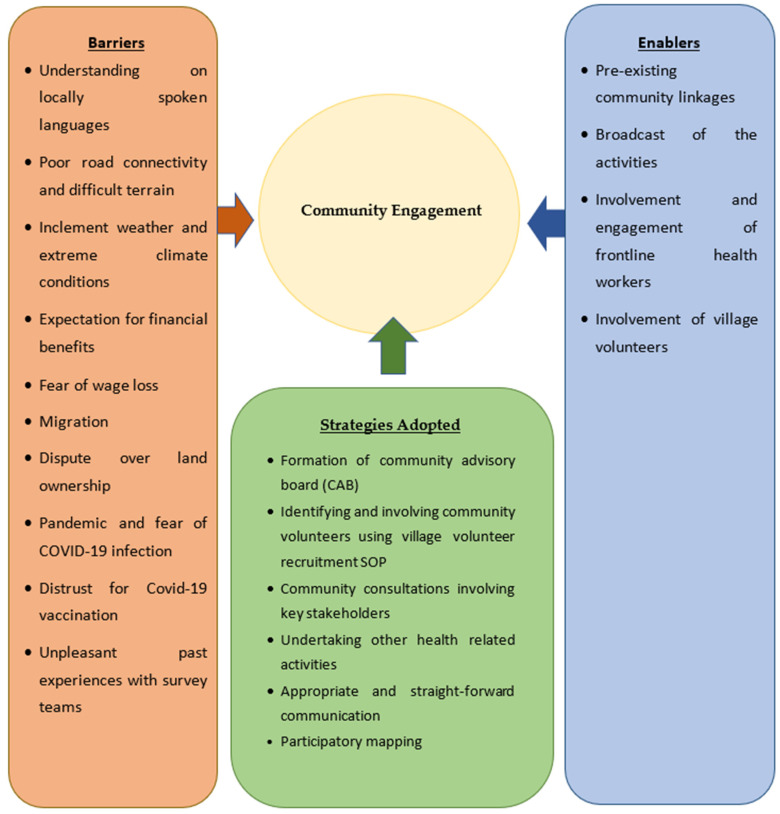
Conceptual framework showing barriers, enablers, and strategies for enhanced community engagement.

## Data Availability

No additional data are available.

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
