# Peer review of "Community Dynamics and Engagement Strategies in Establishing Demographic Development and Environmental Surveillance Systems: A Multi-Site Report from India"

_healthcare, 2023, doi:10.3390/healthcare11030411_

Round 1

Reviewer 1 Report

The manuscript is a very interesting. However, some concerns should be resolved before the next step.

1. The quality of figures are very low. It should be improved.

2. Please describe in "Methods section" a sample size calculation, participant selection strategy, the number of the participants in the cross-sectional study and in the interview stages.

3. Please avoid to use non-classical visualisation in the tables like "X" and "Ö". 

4. The number of participants should be added to the headings of the tables.

5. The main text of the "Results section" should be re-organised and re-written more clear. Because the part related to the interview described like a narrative. Please have a look on other published manuscripts with a qualitative design.

Author Response

We thank the reviewer for their time and constructive comments that have greatly helped us to improve the manuscript. Changes have been done in the manuscript that are highlighted and a point by point response is also attached in the file below. 

Reviewer 2 Report

The proposed manuscript 'Community dynamics and engagement strategies in establishing Demographic Development And Environmental Surveillance Systems: a multi-site report from India' discusses the implementation of public surveillance centers for the assessment of population's health conditions, considering six sites spread across the territory of India. The study identifies most important barriers and enablers, focusing on the engagement of communities, with a specific assessment of the implemented strategies to facilitate such engagement. Despite the low novelty of the proposed methodology, the real-world setting and the extension of the considered projects (both in time and in space) make this study, in my opinion, valuable and of interest to readers. However, there are some issues that I deem to be requiring attention:

MAJOR

1) It is unclear how data were collected. Authors simply report that <<Baseline data was collected from the ongoing multi-site DDESS project in India.>> I understand that, given the complexity of the projects and the diversity of the contexts, a complete and exhaustive description of the data collection process may be too extensive for a single publication; however, some more details are necessary. I suggest adding a paragraph in the methods section dedicated to the structure of the informative flow, the collection process and the analysis of data, while a more extensive description (or least a listing) of the single data sources could be added in the supplementary material.

2) In the discussion part, there is no assessment of the inter-sites differences. In my opinion, an additional paragraph should be added, where authors analyze the differences in barriers, enablers and CE facilitators among the various implementations sites, selecting the most relevant and interesting ones, and trying to interpret them in light of the different characteristics of the areas (as presented in the supplementary material).

MINOR

3) Line 19-20 abstract background: please add a sentence referencing to the research context of the proposed study (state-of-art and relevance of the topic).

4) Line 73 Figure 1: please include in the manuscript a high resolution version of figure 1, which is currently unreadable due to poor image resolution.

5) Line 111-152 methods: when describing the five implementation stages, the manuscript is a little hard to follow. In particular, it is tricky to identify the activities characterizing each stage and the differences among them. I suggest a revision of this section to make it less verbose and more schematic, giving the same structure to the description of each stage.

6) Line 154-196: In my opinion, this section should be moved to the methods part, as it describes the study setting (line 158-196) and the analysis structure (154-157).

7) Line 253: please graphically enlighten the passage from barriers to facilitators analysis.

8) Line 256-264: three enablers are presented in table 2, but four are discussed, where point 2 and 4 of the manuscript both refer to the third row of the table, and the first row of the table is not discussed. Please adjust this inconsistency.

9) Line 268 table 3: it is unclear whether a ‘No’ in the table means that the specific strategy was not implemented in the site, an implementation was attempted but failed, or it was implemented but was unsuccessful in facilitating community engagement. Could authors provide additional details about this aspect?

10) Line 306: the reference to <<epidemiological surveillance>> seems to indicate that it is the sole (or at least the primary) scope of the DDESS implementation, while in my understanding it has a much wider range of aims. I suggest modifying this statement, for example with ‘health conditions surveillance’.

11) I advise a language check to verify the possibility to address some unusual choices, such as line 141 ‘cum’, line 111 ‘rapport’; in general, I suggest using a more common and simple language through the manuscript.

Author Response

(The authors gave the same response as above.)

Round 2

Reviewer 1 Report

Thank you for the significant changes. The manuscript looks well now.